# Structural and Functional Outcomes in Rheumatoid Arthritis After 10-Year Therapy with Disease-Modifying Antirheumatic Drugs Under Tight Control: Evidence from Real-World Cohort Data

**DOI:** 10.3390/jcm14196832

**Published:** 2025-09-26

**Authors:** Shunsuke Mori, Akitomo Okada, Toshimasa Shimizu, Ayuko Takatani, Tomohiro Koga

**Affiliations:** 1Department of Rheumatology, Clinical Research Center for Rheumatic Diseases, National Hospital Organization Kumamoto Saishun Medical Center, Kohshi 861-1196, Japan; 2Department of Rheumatology, National Hospital Organization Nagasaki Medical Center, Omura 856-8562, Japan; akitomoo@hotmail.com; 3Department of Immunology and Rheumatology, Nagasaki University Graduate School of Biomedical Sciences, Nagasaki 852-8523, Japan; toshimasashimizu2000@yahoo.co.jp (T.S.); ayuko.takatani@nagasaki-u.ac.jp (A.T.); tkoga@nagasaki-u.ac.jp (T.K.); 4Clinical Research Center, Nagasaki University Hospital, Nagasaki 852-8501, Japan; 5Department of Public Health, Nagasaki University Graduate School of Biomedical Sciences, Nagasaki 852-8523, Japan

**Keywords:** rheumatoid arthritis, structural remission, functional remission, joint space narrowing, tight control, predictive factors, long-term outcomes

## Abstract

**Objectives**: To examine long-term outcomes and predictors of structural and functional remission in rheumatoid arthritis (RA) after 10-year disease-modifying antirheumatic drug (DMARD) therapy under tight control. **Methods**: We used real-world cohort data from RA patients who completed 10-year DMARD therapy toward remission or low disease activity based on every-3-month measurements between April 2001 and July 2024. Baseline characteristics, disease control during follow-up, and outcomes after 10 years were examined. **Results**: Among 204 patients, 76% received biological and/or non-biological targeted DMARDs. Clinical remission, structural remission defined as an increase in modified total Sharp score (mTSS) ≤ 5 per 10 years, and functional remission defined as health assessment questionnaire-disability index (HAQ-DI) ≤ 0.5 were achieved by 68.1%, 73.0%, and 81.4% of patients, respectively. The mean increase (∆) in mTSS was 5.4 for 10 years (∆erosion score, 1.2; ∆joint space narrowing [JSN] score, 4.2), and 28.9% of patients had no structural progression (51% for erosion score and 34.8% for JSN score). Mean HAQ-DI was 0.26. During a 10-year follow-up, 8.8% of patients experienced high or moderate disease activity lasting for ≥12 months and they had a low structural remission rate (11.1%) and functional remission rate (16.6%). According to multivariable logistic regression analysis, baseline mTSS and JNS score (but not erosion score) were strong predictors for structural and functional remission after 10 years. **Conclusions**: Structural damage progression and functional loss are limited during 10-year tightly controlled DMARD therapy. Compared with bone erosion, JSN appears to be of much higher relevance to structural and functional outcomes.

## 1. Introduction

Rheumatoid arthritis (RA) is a chronic autoimmune disease characterized by synovial membrane inflammation, which causes progressive joint damage [1,2]. Joint damage is closely related to physical function in RA patients and health-related quality of life, especially as disease duration increases [3,4,5]. Systematic reviews of the literature between 1986 and 2001 showed that structural damage, which comprises bone erosion and cartilage degradation, starts early in the disease course and constantly progresses over the first 20 years [3,4]. However, management of RA has been revolutionized over the past decades. The availability of conventional, biological, and non-biological targeted disease-modifying antirheumatic drugs (DMARDs) has dramatically improved long-term outcomes. Early and aggressive intervention of DMARDs with increased treatment choice can induce the prompt suppression of inflammatory disease activity and a significant reduction in joint damage in RA patients [6,7,8].

Tight control of RA is a treatment strategy tailored to individual patients with the aim of achieving a predefined level of disease activity within a certain period of time. This treatment strategy is attained by careful and regular monitoring of disease activity as well as early therapeutic adjustments or switches of DMARD therapies that fail to control disease activity [9,10]. The development and widespread adoption of validated composite measures, which include joint assessment, have allowed disease activity to be quantified reliably [11,12]. Previous clinical trials have indicated that the tight-control strategy can not only adequately control clinical disease activity, but also substantially reduce radiographic progression compared with conventional approaches to the management of RA [13,14,15,16]. In these clinical trials, the treatment goal was clinical remission or low disease activity; measurements of disease activity were performed every 1 to 3 months and, until the desired treatment goal was reached, DMARD therapies were adjusted at least every 3 months [13,14,15,16]. The treat-to-target strategy, which was first developed in 2010 as the guideline principle for the treatment of RA, recommends that treatment be directed to reach and maintain remission or at least low disease activity, and be adjusted toward such a goal at least every 3 months based on the tight-control strategy [17,18]. A recent systematic review and meta-analysis of the literature between 1990 and 2023 indicated that the treat-to-target strategy has significant advantages in increasing clinical response and remission rates and improving health-related quality of life in RA patients [19]. However, there is limited information regarding long-term protective effects of the treat-to-target strategy under tight control on structural damage progression and functional loss, as well as predictive factors for structural and functional remission in real-world settings, especially after various biological and non-biological targeted DMARDs have become available in RA treatment.

Our objectives were to examine long-term outcomes and predictors for structural and functional remission in RA patients after 10-year DMARD therapy under tight control toward remission or at least low disease activity in daily practice. For this study, we conducted a retrospective, long-term follow-up study using real-world cohort data from RA patients who had completed 10-year tight-control DMARD therapy based on every-3-month measurements between April 2001 and July 2024. We examined the contribution of bone erosion and cartilage degradation to structural damage progression and functional loss at 10 years. Additionally, structural and functional outcomes were compared between patients with experience of high or moderate disease activity lasting ≥12 months and those with good disease control during 10-year DMARD therapy.

## 2. Materials and Methods

### 2.1. Patients

We used a real-world database of RA patients who visited the rheumatology division of the National Hospital Organization (NHO) Kumamoto Saishun Medical Center as of April 2001. From this database, we identified patients who started DMARD therapy between April 2001 and July 2014 and had been followed for ≥10 years by July 2024. All participants in this study were required to fulfill the 2010 American College of Rheumatology (ACR)/European Alliance of Associations for Rheumatology (EULAR, formerly the European League Against Rheumatism) criteria for diagnosis of RA [20] and were required to be 18 years of age or older. Patient enrollment flow diagram is shown in Figure 1. We excluded patients if they had any of the following characteristics at their first visit: (1) Steinbrocker radiological stage IV (because of the difficulty in precise scoring of joint damage); (2) previous use of biological DMARD (bDMARD) or targeted synthetic DMARD (tsDMARD); and (3) lack of baseline x-rays of hands and/or feet. We also excluded patients who had failed to receive 10-year DMARD therapy under tight control of clinical disease activity due to the following reasons: (1) discontinuation of DMARD therapy (adverse events or patient’s preference) or (2) drop-out of tight control of clinical disease activity. We did not exclude patients who discontinued DMARD therapy temporarily for certain reasons (e.g., liver enzyme elevations, infection, surgical interventions) and then restarted within 1 month.

Tight control was defined as a treatment strategy in which (1) assessment of clinical disease activity should be performed at each scheduled visit (every 1–3 months) or at any time if patients had aggravation of their clinical signs or symptoms; and (2) DMARD therapy should be modified or changed if patients cannot achieve remission or at least low disease activity within 3 months or if patients suffer from a disease flare [9,10]. A disease flare was defined as a worsening of the disease activity index (a return to moderate or high activity) at any time during follow-up.

### 2.2. Data Collection at Baseline

We reviewed our patient records and collected clinical data at the time of their first visits (baseline characteristics), which included demographic characteristics (age and sex), RA-related factors (28-joint disease activity score based on erythrocyte sedimentation rate [DAS28-ESR], duration of joint signs and/or symptoms (disease duration), anti-cyclic citrullinated peptide antibodies [anti-CCP], and rheumatoid factor [RF]), smoking history, and body mass index (BMI). Sera that had been collected at the first visit and stored at −80 °C were used for the detection of anti-CCP and RF. Detailed methods for the measurement have been described in our previous study [21]. The previous use of conventional synthetic DMARDs (csDMARDs) and year of the patient’s first visit at our institution were also recorded. The availability of DMARDs differed between 2001 and 2014. To explore the possibility that this difference in DMARD availability during the early stage of a patient’s treatment might affect structural and functional outcomes after 10 years, we included the year of each patient’s first visit in the baseline characteristics.

### 2.3. Monitoring Clinical Disease Activity During Follow-Up

For monitoring disease activity, we used DAS28-ESR between 2001 and 2005. Since 2006, the clinical disease activity index (CDAI) has been used to follow patients [11,22]. For CDAI, cut-off values for disease activity states were defined as follows: >22 for high disease activity; >10 and ≤22 for moderate disease activity; >2.8 and ≤10 for low disease activity; and ≤2.8 for remission [11]. For DAS28-ESR, the following definition was used: >5.1 for high disease activity; ≥3.2 and ≤5.1 for moderate disease activity; ≥2.6 and <3.2 for low disease activity; and <2.6 for remission [23]. For all patients, the disease activity state after 10-year DMARD therapy was determined based on CDAI values.

Based on the clinical disease activity during the 10-year follow-up, each RA patient was classified into a poor control group (patients with experience of high or moderate disease activity lasting for ≥12 months), a moderate control group (patients with experience of high or moderate disease activity lasting for 3–12 months), and a good control group (patients without experience of high or moderate disease activity lasting for ≥3 months).

### 2.4. Assessment of Joint Destruction

To assess structural damage, we use radiographic imaging of each patient’s hands and feet taken both at baseline and after 10 years of DMARD therapy. Each radiograph was assessed independently by two rheumatologists who were well trained and competent to score radiographs using the van der Heijde-modified total Sharp score (mTSS) [24,25]. The readers were blinded to the patient’s clinical status and treatment. To evaluate structural damage progression, a change in mTSS from baseline to 10 years later (∆mTSS) was calculated for each patient. The mTSS is the sum of the erosion score (range 0–280) and the joint space narrowing (JSN) score (range 0–168). In addition to mTSS, we also reported erosion score and JSN score separately because these scores can provide complementary information on different aspects of structural damage [26,27]. The inter-observer agreement between the two readers was determined using the intraclass correlation coefficient (ICC). A two-way mixed-effects model with consistency definition and single rater type (ICC (3,1)) was applied [28]. The ICC (3,1) for ∆mTSS, erosion score, and JNS score were 0.907, 0.953, and 0.963, respectively, which indicated excellent agreement in radiographic scoring.

### 2.5. Outcomes of Interest

The outcomes of interest were structural changes defined as the ∆mTSS, functional impairment expressed by the health assessment questionnaire-disability index (HAQ-DI), and improvement of clinical disease activity after 10 years of DMARD therapy under tight control. Structural and functional remission were defined as ∆mTSS ≤ 5.0 per 10 years and HAQ-DI ≤ 0.5 at 10 years, respectively. Clinical remission was defined as CDAI ≤ 2.8 at 10 years.

### 2.6. Statistical Analysis

Mean and standard deviation (SD) were used as descriptive statistics for data with a continuous distribution, which included non-normally distributed data [29]. Number (percentage) was used for categorical data. We compared baseline characteristics and DMARD use between two patient groups using the independent-measures *t*-test for continuous variables and Fisher’s exact probability test for categorical variables. Comparisons of outcomes between three patient groups were performed using a one-way analysis of variance (ANOVA) with post hoc Tukey’s honestly significant difference (HSD) test for continuous variables and using Fisher’s exact probability test with the post hoc Holm test for categorical variables. There were no missing measurements at baseline, clinical disease activity state and DMARD use during follow-up, or outcomes after 10 years.

Logistic regression analysis was performed to evaluate the association between structural or functional remission after 10-year DMARD therapy as a dependent variable and a set of baseline characteristics as independent variables that were considered to be clinically relevant based on previous knowledge, which included age, sex, RA-related factors, smoking history, BMI, previous use of csDMARDs, and year of the first visit. Additionally, mTSS (erosion score/JSN score) at baseline was introduced into this analysis as an independent variable. Univariable logistic regression analysis was performed first for each independent variable. Thereafter, all variables with *p*-values < 0.10 in the univariable models were introduced into multivariable logistic regression analysis. The mTSS and erosion score/JSN score were included separately in multivariable regression analysis (Model 1 for mTSS and Model 2 for erosion score/JSN score), because erosion score and JSN score are compositions of mTSS. A forced entry procedure was used in the multivariable model. The strength of association between structural or functional remission and the independent variables was estimated using odds ratios (ORs) and 95% confidence intervals (95% CIs). Receiver operating characteristic (ROC) curves and the corresponding area under the curve (AUC) values were calculated to provide an index of validity for the multivariable logistic regression model.

For all tests, *p*-values < 0.05 were considered to indicate statistical significance. All calculations were performed using PASW Statistics version 27 (SPSS Japan Inc., Tokyo, Japan) and Easy R (Saitama Medical Center, Jichi Medical University, Saitama, Japan) [30].

## 3. Results

### 3.1. Baseline Characteristics of RA Patients Grouped by Achievement of Structural and Functional Remission After 10-Year DMARD Therapy

As shown in Figure 1, we identified 204 patients with RA (50 males and 154 females) who completed 10-year DMARD therapy under tight control. Baseline characteristics are shown in Table 1. The mean age at baseline was 57.5 years, and the mean disease duration was 1.8 years. More than 80% of patients did not have previous use of any type of DMARDs at baseline. The mean DAS28-ESR was 4.8. The mean mTSS at baseline was 8.4 (erosion score, 3.2; JSN score, 5.2) and 30.9% had normal joint space without bone erosion at baseline (mTSS = 0).

After 10-year DMARD therapy under tight control, 195 patients (95.6%) achieved clinical remission (68.1% for CDAI ≤ 2.8) or low disease activity (27.5% for CDAI > 2.8 and ≤10). In total, 149 patients (73.0%) in structural remission and 166 patients (81.4%) were in functional remission (Figure 1). As shown in Table 1, mean mTSS, erosion score, and JNS score at baseline were significantly lower in the structure remission and functional remission groups than in the non-remission groups (structural remission: 5.8 versus 15.5 for mTSS, 2.4 versus 5.5 for erosion score, 3.5 versus 10.0 for JSN score, *p* < 0.001; functional remission: 6.6 versus 16.5 for mTSS, 2.6 versus 6.1 for erosion score, 4.0 versus 10.4, for JSN score, *p* < 0.001). Regarding demographic characteristics, patients in the functional remission group were significantly younger compared with those in the non-remission group (56.7 versus 61.3 years, *p* = 0.013). Rates of male patients were higher in the structural remission and functional remission groups than in the non-remission groups (structural remission: 28.9% versus 12.7%, *p* = 0.018; functional remission: 27.4% versus 11.1%, *p* = 0.035). Regarding RA-related characteristics, the mean DAS28-ESR values were lower in the structural remission and functional remission groups compared with the non-remission groups (structural remission: 4.7 versus 5.1, *p* = 0.030; functional remission: 4.7 versus 5.4, *p* < 0.001). Disease duration was significantly shorter in the structural remission group than in the non-remission group (1.4 years versus 2.8 years, *p* = 0.020).

### 3.2. Clinical, Functional, and Structural Outcomes in RA Patients Grouped by Disease Activity Control During 10-Year DMARD Therapy

Detailed clinical, functional, and structural outcomes after 10 years are shown in Table 2. Mean CDAI and HAQ-DI were 2.8 and 0.26, respectively. Mean mTSS was 13.8 (erosion score, 4.4; JSN score, 9.4). Mean increase in mTSS (∆mTSS) for 10 years was 5.4 (∆erosion score, 1.2; ∆JSN score, 4.2). Median ∆mTSS was 2.0. Under tight control, 28.9% of patients had no structural changes for 10 years, 51.0% had no progression of bone erosion, and 34.8% had no changes in JSN (Table 2 and Figure 2). The contribution of JSN changes to joint damage progression was much higher compared with progression of erosive changes.

We classified RA patients into three categories; that is, good, moderate, and poor control groups, based on the experience of high or moderate disease activity for 10 years (Table 2). In total, 134 patients (65.7%) had no experience of high or moderate disease activity lasting for ≥3 months (good control group). In contrast, 18 patients (8.8%) experienced high or moderate disease activity lasting for ≥12 months (poor control group), and 52 patients (25.5%) had high or moderate disease activity lasting for 3–12 months (moderate control group). Ten years later, the means of CDAI, HAQ-DI, mTSS, and ∆mTSS, as well as the rates of patients with clinical, functional, and structural remission, were significantly different among the three patient groups. In the good control group, more than 90% of patients were in structural remission and functional remission at 10 years. In contrast, the rates of structural remission and functional remission in the poor control group were 11.1% and 16.6%, respectively. Progression of bone erosion and JSN was significantly slower in the good control group compared with the poor and moderate control groups, but their contributions to structural changes were similar among the three control groups.

### 3.3. DMARD Use for 10 Years in RA Patients Under Tight Control

All patients, except those with contraindication for methotrexate (MTX), started DMARD therapy with MTX monotherapy. As shown in Table 3, 96.6% of patients used MTX. MTX exposure was significantly longer in the structural remission and functional remission groups than in the non-remission groups (structural remission: 109.1 versus 92.0 months; functional remission: 107.4 versus 89.7 months, *p* < 0.001). In total, 155 patients (76%) received bDMARD and/or tsDMARD. The rate of bDMARD/tsDMARD use for 10 years was significantly lower in the functional remission group compared with the non-remission group (71.7% versus 94.7%, *p* = 0.001). A similar trend was observed in a comparison between the structural remission group and the non-remission group (72.5% versus 85.5%, *p* = 0.065). The number of bDMARD/tsDMARD classes used per patient for 10 years was significantly smaller in the structural remission and functional remission groups compared with the non-remission groups (structural remission: 1.2 versus 1.9, *p* < 0.001; functional remission: 1.2 versus 2.3, *p* < 0.001). The number of failures per patient was significantly smaller in the structural remission and functional remission groups than in the non-remission groups (structural remission: 0.5 versus 1.2, *p* < 0.001; functional remission: 0.5 versus 1.5, *p* < 0.001). Thus, RA patients in the non-remission groups received more classes of bDMARD/tsDMARD and experienced more failures during 10-year DMARD therapy compared with those in the remission groups.

### 3.4. Predictors of Structural Remission After 10-Year DMARD Therapy Under Tight Control

Univariable logistic regression analyses revealed that male sex, DAS28-ESR, disease duration, mTSS, erosion score, and JSN score at baseline were significantly associated with structural remission after 10 years (Table 4). Baseline mTSS and other baseline characteristics with *p* < 0.10 in the univariable analyses were introduced into the multivariable analysis as independent variables (Model 1), which showed that baseline mTSS was the only variable to predict structural remission after 10 years. Adjusted ORs (95% CIs) for baseline mTSS were 0.15 (0.05–0.46) for >5 and ≤25 (*p* < 0.001) and 0.12 (0.03–0.48) for >25 (*p* = 0.003) versus 0 (normal joint space without erosion at baseline). When baseline erosion score and JSN score, instead of mTSS, were introduced into the multivariable analysis as independent variables (Model 2), the JSN score and the first visit year were identified as variables for predicting structural remission after 10 years. Adjusted ORs (95% CIs) for JSN score were 0.29 (0.10–0.83) for >3 and ≤10 (*p* = 0.021) and 0.14 (0.04–0.47) for >10 (*p* = 0.002) versus 0 (normal joint space). Adjusted OR (95% CI) for the year of the first visit was 2.45 (1.04–5.77) for 2009–2011 versus 2001–2008 (*p* = 0.040).

### 3.5. Predictors of Functional Remission After 10-Year DMARD Therapy Under Tight Control

In univariable logistic regression analyses, we identified advanced age, male sex, DAS28-ESR, mTSS, erosion score, and JSN score at baseline as variables significantly associated with functional remission after 10 years (Table 5). When baseline mTSS and other baseline characteristics with *p* < 0.10 in the univariable analyses were introduced into the multivariable analysis as independent variables (Model 1), mTSS, advanced age, male sex, and DAS28-ESR were identified as predictor variables of functional remission at 10 years. Adjusted ORs (95% CIs) for baseline mTSS were 0.17 (0.04–0.70) for >0 and ≤5 (*p* = 0.014), 0.10 (0.02–0.43) for >5 and ≤10 (*p* = 0.002), and 0.07 (0.01–0.34) for >10 (*p* = 0.001) versus 0 (norm joint space without erosion at baseline). Adjusted ORs (95% CIs) for age > 65 years, male sex, and DAS28-ESR per additional unit were 0.29 (0.12–0.70) (*p* = 0.006), 6.66 (1.91–23.15) (*p* = 0.003), and 0.55 (0.38–0.80) (*p* = 0.002), respectively. When we introduced baseline erosion score and JSN score, instead of mTSS, into multivariable analysis as independent variables (Model 2), JSN score, advanced age, male sex, and DAS28-ESR were detected as variables to predict functional remission after 10 years. Adjusted ORs (95% CIs) for baseline JSN score were 0.17 (0.05–0.58) for >3 and ≤10 (*p* = 0.005) and 0.09 (0.02–0.38) for >10 (*p* < 0.001) versus 0 (normal joint space). Adjusted ORs (95% CIs) for age > 65 years, male sex, and for DAS28-ESR per additional unit was 0.27 (0.10–0.69) (*p* = 0.006), 6.15 (1.66–22.74) (*p* = 0.006), and 0.57 (0.39–0.85) (*p* = 0.006), respectively.

### 3.6. Sensitivity Analysis

Considering the influence of disease control for the 10-year follow-up on structural and functional outcomes, we performed univariable and multivariable logistic regression analyses for RA patients who experienced high or moderate disease activity lasting for ≥3 months (the poor and moderate control groups) (Appendix A). In multivariable logistic regression models, baseline mTSS (Model 1) and JSN score (Model 2) were identified as the strong predictors of structural and functional remission after 10 years.

## 4. Discussion

In the present study, 76% of RA patients were treated with bDMARDs and/or tsDMARDs. Clinical, structural, and functional remission were achieved in 68.1%, 73.0%, and 81.4%, respectively, of RA patients who completed 10-year DMARD therapy under tight control. Mean ∆mTSS per 10 years was 5.4 (∆erosion score, 1.2; ∆JNS score, 4.2), and 28.9% of patients had no structural progression for 10 years (51% without erosive progression and 34.8% without JNS progression). Mean HAQ-DI was 0.26. During 10-year DMARD therapy, 65.7% of patients showed good control of disease activity, but 8.8% experienced high or moderate disease activity lasting for ≥12 months. The disease control significantly affected structural and functional outcomes after 10 years. For all patients, baseline mTSS and JSN score (but not erosion score) were the strong predictors for structural and functional remission after 10 years. Even for patients who failed at good disease control during follow-up, baseline mTSS and JSN were identified as factors predicting structural and functional remission.

Radiographic progression and functional loss in the present study were limited compared with previous 10-year follow-up studies without tight control. A prospective cohort study for RA patients recruiting from 1993 to 1994 reported that the mean mTSS of hands and feet increased to 35.4 at 10 years from 5.8 at baseline (average annual progression rate, 2.96/year) [31]. Longitudinal follow-up studies for RA patients starting in 1992 (EURIDISS cohort) showed that the mean mTSS of hands (scale 0–280) worsened to 36.0 at 10 years from 6.8 at baseline (mean yearly progression rate, 2.8). Mean HAQ-DI at 10 years was 0.92 [32,33,34]. In a 10-year follow-up study of Japanese RA patients recruiting in 1995, the mean mTSS worsened from 5 to 35 after 10 years [35]. Most patients in those studies were treated with MTX and other csDMARDs without tight control during follow-up. In a systematic review and meta-analysis using data from longitudinal observational cohorts (follow-up range, 5–20 years), Carpenter et al. showed that the annual progression rate of structural joint damage in studies recruiting between 1990 and 2011 was significantly lower compared with that in studies recruiting between 1965 and 1989 because of critical changes in treatment practices [36].

Previous clinical trials have shown that the treat-to-target strategy with MTX monotherapy or combination with infliximab or adalimumab induced long-term protective effects on structural damage progression and functional loss in RA patients. The BeSt study compared four different treatment strategies, namely arms 1 and 2 starting with MTX monotherapy (switching to or adding other DMARD) and arms 3 and 4 starting with MTX combination therapy (prednisolone or infliximab) in patients with early RA. Those patients were recruited between 2000 and 2002 and received tightly controlled targeted treatment to achieve low disease activity [14]. For all strategies, subsequent treatment adjustments were done based on DAS44 measurements every 3 months [37]. Over 10 years, 82% had low disease activity (DAS44 ≤ 2.4) and 53% were in remission (DAS44 < 1.6). Mean estimated ∆mTSS during follow-up was 10.9, 8.4, 8.1, and 6.1 (median 2.0, 2.5, 3.0, and 1.5) for arms 1–4, respectively. Mean HAQ-DI after 10 years was 0.69, 0.72, 0.64, and 0.58 in arms 1–4, respectively. All treat-to-target strategies were equally effective in preventing functional disability and radiographic progression after 10 years [37,38]. The IMPROVED study determined 5-year outcomes of early remission induction therapy with MTX plus prednisolone followed by targeted treatment aiming at drug-free remission (DAS44 < 1.6) with a combination of MTX and other csDMARD (arm 1) and a combination of MT and adalimumab (arm 2) in early RA patients who were included between 2007 and 2010. Tight control was done based on DAS44 measures every 4 months [39,40]. After 5 years, 48% were in remission. Mean ∆mTSS was 2.0 (median 0.5) and 1.7 (median 0.29) in arms 1 and 2, respectively. Mean HAQ-DI was 0.83 and 0.82 in arms 1 and 2, respectively. There were no significant differences between the randomization arms [41].

In the 8-year open-label extension of PREMIER trials, patients with early RA who had completed an initial double-blinded clinical trial with adalimumab plus MTX (arm 1), adalimumab alone, (arm 2), and MTX alone (arm 3) for 2 years were then followed by adalimumab with and without MTX during the open-label extension for 8 years. For 10-year completers, patients in the arm 1 achieved remission more readily compared with arms 2 and 3 (75.6% in arm 1, 61.7% in arm 2, and 56.2% in arm 3). Patients initially randomized to arm 1 displayed better structural and functional outcomes after 10 years, particularly in the prevention of radiographic progression; mean ∆mTSS was 4.0, 8.8, and 11.0 for arms 1–3, respectively. Mean HAQ-DI was 0.4, 0.7, and 0.6 in arms 1–3, respectively. Initial intensive therapy in patients with early RA had long-term benefits, which persisted up to 10 years [42]. Similar results were obtained in the 9-year open-label extension of the DE019 study (a 1-year randomized controlled trial for adalimumab in patients with long-standing RA) [43].

In the present analysis of real-world cohort data, similar levels of annual mean ∆mTSS rate and mean HAQ-DI were observed to the above-mentioned results from the BeSt, IMPROVED, and PREMIER studies. Radiographic progression was well suppressed, and good functional ability was preserved after 10-year DMARD therapy under tight control. The majority of our patients started RA treatment with MTX monotherapy, and if the treatment target was not achieved within 3 months, MTX monotherapy was switched to bDMARD or tsDMARD therapy with or without MTX. For 10 years, 76% of patients received bDMARDs and/or tsDMARDs with and without MTX. Thus, tightly controlled DMARD therapy from the start of RA treatment can lead to favorable long-term structural and functional outcomes in RA patients after 10 years in the real-world setting.

In the present study, structural and functional outcomes were worse in RA patients who experienced high or moderate disease activity lasting for ≥12 months during 10-year DMARD therapy compared with those who maintained good disease activity control. In the ESPOIR cohort study recruiting between 2002 and 2005, 10-year radiographic and functional outcomes were better in patients in sustained remission compared with those in sustained low disease activity and those in sustained moderate or high disease activity [44]. In a post hoc analysis of 10-year results from the BeSt study, more RA patients who had initially received MTX combination therapy achieved early and continuous low disease activity than those who had started with MTX monotherapy. Regardless of initial therapy, however, RA patients with continuous low disease activity had similar long-term clinical and radiological outcomes [45]. In another post hoc analysis of 10-year BeSt results, disease flares were associated with long-term joint damage progression and functional deterioration with a dose–response relationship with the number of flares [46]. These results indicated that good control of clinical disease activity during a 10-year follow-up is important in achieving structural and functional remission. In the present study, RA patients in the non-remission groups received more classes of bDMARD/tsDMARD and experienced more failures during 10-year DMARD therapy compared with those in the structural remission and functional remission groups. The rate of users of ≥3 bDMARD/tsDMARD classes was significantly higher in the non-remission groups than in the remission groups. The users of ≥3 classes had 2 or more switches of a current bDMARD/tsDMARD to another one mainly because of the need for treatment adjustments to control clinical disease activity. In other words, these patients might experience high or moderate disease activity for a longer time than users of 1 or 2 bDMARD/tsDMARD classes. The data suggest that sustained remission or at least low disease activity during follow-up is important in achieving structural and functional remission after 10 years. In the present study, we assessed clinical disease activity every 1 to 3 months and, if at least low disease activity was not achieved within 3 months, DMARD therapy was modified or changed. However, more frequent monitoring and earlier switching or adjustments of DMARD therapy may be required for patients with high or moderate disease activity.

We showed that baseline mTSS was a strong predictor of structural remission at the end of a 10-year DMARD therapy under tight control. Ten-year cohort studies in Europe also showed that baseline radiographic damage in RA patients was predictive of long-term radiographic joint damage, although most patients in those studies were treated with csDMARDs [31,47]. In the post hoc analysis of patients with early RA in the ASPIRE study, the progression of bone erosions and JSN was compared between MTX monotherapy and infliximab plus MTX combination therapy. Among patients with development or progression of joint damage after 54 weeks, bone erosion occurred more often than JNS in both treatment groups, suggesting that progression of bone erosion and JSN are potentially independent events. In addition, there was a tendency for joints with existing erosions or JSN at baseline to have progression of the same type of damage, rather than development of new damage [48]. Long-term effects of bone erosion and JSN on joint damage progression were reported in the long-term extension of BeSt, IMPROVED, and DE019 trials, in which the JSN component (cartilage damage) appeared to drive the radiological progression compared with bone erosions regardless of initial treatment [43,49]. In the present study, baseline JSN score, but not erosion score, was significantly associated with future structural outcomes, and JNS was the predominant type of joint damage after 10 years of DMARD therapy. Cartilage destruction seems to be of much higher relevance to joint damage progression than bone destruction.

Physical dysfunction has been reported as a consequence of ongoing inflammatory activity and structural joint damage through the course of RA; in early RA, it is most associated with the extent of disease activity, and in late RA, with structural damage [50,51]. Physical dysfunction is partly reversible, which is mediated by disease activity. Reversibility of physical disability decreases with the duration of RA, and in the absence of disease activity, structural damage contributes to irreversible physical dysfunction [52]. A relationship between structural damage and impaired physical function was also shown in the EURIDISS 10-year follow-up studies. Structural damage and disease activity are independent contributors to impaired physical function early and late in the RA process. [32,34]. Using data from several clinical trials for TNF inhibitors in RA patients, Aletaha et al. showed that cartilage damage was more clearly associated with irreversible physical disability at the time of remission than bone damage [53]. We found that baseline JSN score (but not erosion score) was strongly associated with not only structural damage but also physical dysfunction after 10-year DAMRD therapy. In addition to JSN score, baseline DAS28-ESR was also a strong predictor for physical function after 10 years. High DAS28-ESR had a trend to contribute to poor structural outcomes, but this association was not statistically significant in the present study.

In previous 10-year follow-up studies, high levels of disease activity (especially exemplified by acute phase reactants and swollen joint counts) and the presence of autoantibodies at baseline have been reported as prognostic factors associated with progression of joint damage in RA [31,33,47], but these baseline characteristics were not identified as the predictors for poor structural outcomes in the present study. Using 5-year results from the IMPROVED study, van der Pol et al. showed that, in patients with these poor prognostic factors, there was a non-significant increase in the risk of joint damage progression as the number of these factors increased [54].

There are some limitations to this study. First, this was a long-term retrospective observational study, which may confer inherent limitations. However, the medical records in our database allowed us to obtain accurate clinical data at baseline, information about DMARD use and control status of clinical disease activity during follow-up, and structural and functional outcomes after 10 years. Second, the aim of this study was to evaluate long-term outcomes and identify baseline characteristics predicting structural and functional remission in RA patients who completed 10-year DMARD therapy under tight control. Therefore, we excluded patients who dropped out of tightly controlled DMARD therapy during the 10-year follow-up. However, we showed that, even in patients who completed 10-year tight-control therapy, approximately 8.8% experienced high or moderate disease activity lasting for ≥12 months, and these poorly controlled patients had significantly lower structural and functional remission rates compared with patients who maintained good disease activity control. Third, various types of bDMARDs and tsDMARDs were used with and without MTX during 10-year tight-control therapy. We could not examine the effects of each DMARD therapy on long-term outcomes because of the small number of patients in individual treatment groups and the various exposure times.

## 5. Conclusions

Tightly controlled DMARD therapy toward remission or at least low disease activity can contribute to the suppression of structural damage and physical dysfunction after 10 years in the real-world setting. Baseline mTSS was a strong predictor for structural and functional remission. JSN seems to be more clearly associated with joint damage progression and functional loss than bone erosion. Patients with the experience of high or moderate disease activity lasting for ≥12 months during follow-up are at increased risk of poor structural and functional outcomes. Even for patients who failed at good disease control during follow-up, baseline JSN is a strong factor predicting structural and functional remission. To obtain favorable long-term structural and functional outcomes, particular attention should be given to the prevention of cartilage damage progression during DMARD therapy under tight control in daily practice.

## Figures and Tables

**Figure 1 jcm-14-06832-f001:**
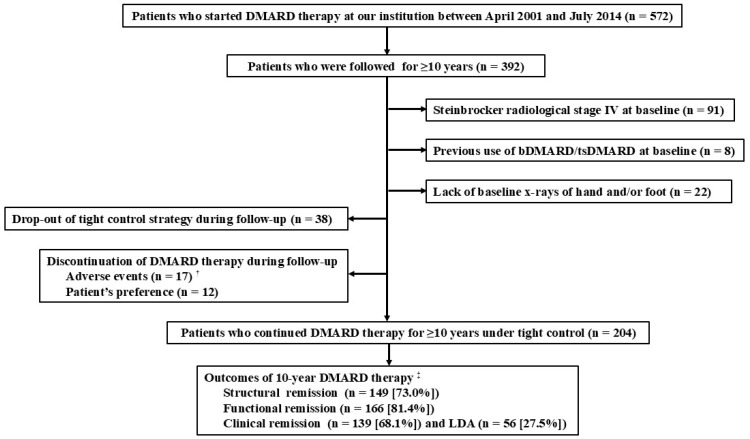
Patient enrollment flow diagram ^†^ Included infection (*n* = 4), fracture (*n* = 4), malignancy (*n* = 3), Fabry disease (*n* = 1), dementia (*n* = 1), de novo hepatitis B virus infection (*n* = 1), cardiac infarction (*n* = 1), chronic pulmonary embolism (*n* = 1), and lymphopenia (*n* = 1). We did not exclude patients who discontinued DMARD therapy temporarily and restarted within 1 month. ^‡^ Defined as ∆mTSS ≤ 5.0 for structural remission, HAQ-DI ≤ 0.5 for functional remission, CDAI ≤ 2.8 for clinical remission, and CDAI > 2.8 and ≤10 for LDA. CDAI clinical disease activity index; DMARD, disease-modifying antirheumatic drug; bDMARD, biological DMARD; tsDMARD, targeted synthetic DMARD; HAQ-DI, health assessment questionnaire-disability index; LDA, low disease activity; mTSS, van der Heijde-modified total Sharp score; RA, rheumatoid arthritis.

**Figure 2 jcm-14-06832-f002:**
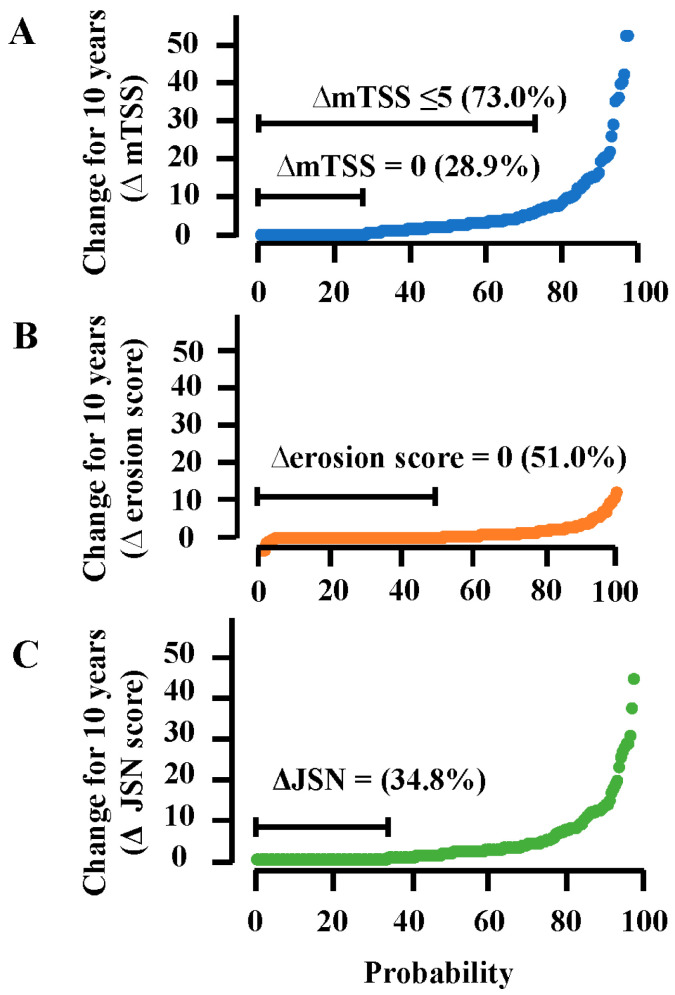
Cumulative probability plots of changes in mTSS (**A**), erosion score (**B**), and JSN score (**C**) for 10 years of DMARD therapy under tight control. JSN score, joint space narrowing score; mTSS, van der Heijde-modified total Sharp score.

**Table 1 jcm-14-06832-t001:** Baseline characteristics of RA patients grouped by achievement of structural and functional remission after 10-year tight-control DMARD therapy.

Baseline Characteristics	Total	Structural Remission ^¶^		*p* *	Functional Remission ^¶^		*p* *
	(*n* = 204)	Yes	No		Yes	No	
		(*n* = 149)	(*n* = 55)		(*n* = 166)	(*n* = 38)	
Age, years, mean (SD)	57.5 (10.4)	57.9 (10.8)	56.4 (9.3)	0.34	56.7 (9.9)	61.3 (11.7)	0.013
>65 years, number (%)	48 (23.5)	38 (25.5)	10 (18.2)	0.35	32 (19.0)	16 (44.4)	0.005
Male, number (%)	50 (24.5)	43 (28.9)	7 (12.7)	0.018	46 (27.4)	4 (11.1)	0.035
Anti-CCP positive, number (%)	179 (87.7)	127 (85.2)	52 (94.5)	0.092	144 (86.7)	35 (92.1)	0.58
RF positive, number (%)	162 (79.4)	114 (76.5)	48 (87.3)	0.12	131 (78.9)	31 (81.6)	0.83
DAS28-ESR, mean (SD)	4.8 (1.2)	4.7 (1.2)	5.1 (1.1)	0.030	4.7 (1.1)	5.4 (1.2)	<0.001
High (>5.1) number (%)	86 (42.2)	58 (38.9)	28 (50.9)	0.15	63 (38.0)	23 (60.5)	0.017
Disease duration ^†^, years, mean (SD)	1.8 (3.5)	1.4 (2.9)	2.8 (4.7)	0.020	1.7 (3.5)	2.4 (3.5)	0.23
≤6 months, number (%)	126 (61.8)	94 (63.1)	32 (58.2)	0.52	106 (63.9)	20 (52.6)	0.20
Year of the first visit **^‡^**, number (%)							
2001–2008	75 (36.8)	58 (38.9)	17 (30.9)	0.33	62 (37.3)	13 (34.2)	0.85
2009–2011	77 (37.7)	55 (36.9)	22 (40.0)	0.75	63 (38.0)	14 (36.8)	1.00
2012–2014	52 (25.5)	36 (24.2)	16 (29.1)	0.47	41 (24.7)	11 (28.9)	0.68
Previous csDMARD use, number (%)							
No use	173 (84.8)	129 (86.6)	44 (80)	0.27	141 (84.9)	32 (84.2)	1.00
Smoking history > 30 PYs, number (%)	31 (15.2)	27 (18.1)	4 (7.3)	0.077	28 (16.9)	3 (7.9)	0.21
BMI > 25, number (%)	32 (15.7)	19 (12.8)	13 (23.6)	0.081	24 (14.5)	8 (21.1)	0.13
mTSS (range 0–448) ^§^, mean (SD)	8.4 (13.7)	5.8 (10.7)	15.5 (18.0)	<0.001	6.6 (11.6)	16.5 (18.7)	<0.001
0, number (%)	63 (30.9)	57 (38.3)	6 (10.9)	<0.001	60 (36.1)	3 (7.9)	<0.001
>0 and ≤5, number (%)	66 (32.4)	50 (33.6)	16 (29.1)	0.62	53 (31.9)	13 (34.2)	0.85
>5 and ≤25, number (%)	55 (27.0)	33 (22.1)	22 (40)	0.013	41 (24.5)	14 (36.8)	0.16
>25, number (%)	20 (9.8)	9 (6.0)	11 (20)	0.006	12 (7.2)	8 (21.1)	0.016
Erosion score, mean (SD)	3.2 (6.0)	2.4 (4.6)	5.5 (8.3)	<0.001	2.6 (4.6)	6.1 (9.6)	<0.001
No erosion, number (%)	94 (46.1)	75 (50.3)	19 (34.5)	0.057	85 (51.2)	9 (23.7)	0.002
>0 and ≤3, number (%)	63 (30.9)	46 (30.9)	17 (30.9)	1.00	46 (27.7)	17 (44.7)	0.051
>3 and ≤10, number (%)	26 (12.7)	19 (12.8)	7 (12.7)	1.00	22 (13.3)	4 (10.5)	0.79
>10, number (%)	21 (10.3)	9 (6.0)	12 (21.8)	0.003	13 (7.8)	8 (21.1)	0.033
Only erosion, number (%)	33 (16.2)	20 (13.4)	13 (23.6)	0.089	28 (16.9)	5 (13.2)	0.79
JSN score, mean (SD)	5.2 (8.7)	3.5 (6.6)	10.0 (11.5)	<0.001	4.0 (7.7)	10.4 (10.9)	<0.001
Normal joint space, number (%)	97 (47.5)	83 (55.7)	14 (25.5)	<0.001	89 (53.6)	8 (21.1)	<0.001
>0 and ≤3, number (%)	33 (16.2)	26 (17.4)	7 (12.7)	0.52	28 (16.9)	5 (13.2)	0.81
>3 and ≤10, number (%)	35 (17.2)	23 (15.4)	12 (21.8)	0.30	27 (16.3)	8 (21.1)	0.48
>10, number (%)	39 (19.1)	17 (11.4)	22 (40)	<0.001	22 (13.3)	17 (44.7)	<0.001
Only JSN, number (%)	36 (17.6)	28 (18.8)	8 (14.5)	0.54	30 (18.1)	6 (15.8)	0.61

* Compared between patients who achieved remission and those who failed in achieving remission, using Fisher’s exact probability test for categorical variables and independent-measures *t*-test for continuous variables. ^†^ Defined as the time length from onset of joint symptoms and/or signs to the first visit at our institution. ^‡^ Available DMARDs: 2001–2008, TNF inhibitors; 2009–2011, plus IL-6 inhibitors and abatacept; 2012–2014, plus JAK inhibitors. ^§^ Determined as the sum of the erosion score (range 0–280) and the JSN score (range 0–168). ^¶^ Defined as ∆mTSS ≤ 5.0 per 10 years for structural remission and HAQ-DI ≤ 0.5 at 10 years for functional remission. anti-CCP, anti-cyclic citrullinated peptide antibodies; BMI, body mass index; csDMARDs, conventional synthetic DMARDs; DAS28-ESR, 28-joint disease activity score using erythrocyte sedimentation rate; DMARD, disease-modifying antirheumatic drug; HAQ-DI, health assessment questionnaire-disability index; IL-6, interleukin-6; JAK, Janus kinase; JSN, joint space narrowing; mTSS, van der Heijde-modified total Sharp score; PYs, pack-years; RA, rheumatoid arthritis; RF, rheumatoid factor; SD, standard deviation; TNF, tumor necrosis factor.

**Table 2 jcm-14-06832-t002:** Outcomes in RA patients grouped by experience of high or moderate disease activity during 10-year tight-control DMARD therapy.

Outcomes at 10 Years	Total	Good Control ^†^	Moderate Control ^†^	Poor Control ^†^	*p* *
	(*n* = 204)	(*n* = 134)	(*n* = 52)	(*n* = 18)	
Clinical disease activity at 10 years					
CDAI, mean (SD)	2.8 (3.1)	1.9 (2.1)	3.5 (3.5)	7.6 (3.6)	<0.001
Remission (<2.8), number (%)	139 (68.1)	109 (81.3)	28 (53.8)	2 (11.1)	<0.001
Low (>2.8 and ≤10), number (%)	56 (27.5)	23 (17.2)	20 (38.5)	13 (72.2)	<0.001
Physical function at 10 years					
HAQ-DI, mean (SD)	0.26 (0.49)	0.12 (0.34)	0.35 (0.45)	1.05 (0.69)	<0.001
HAQ-DI ≤ 0.5 (remission), number (%)	166 (81.4)	126 (94.0)	37 (71.2)	3 (16.6)	<0.001
Structural changes for 10 years					
mTSS at 10 years, mean (SD)	13.8 (17.8)	7.8 (12.0)	22.5 (20.6)	34.1 (20.9)	<0.001
∆mTSS, mean (SD)	5.4 (9.0)	1.7 (2.2)	8.6 (9.3)	22.9 (14.6)	<0.001
∆mTSS, median (IQR)	2.0 (0, 6)	1.0 (0, 3.0)	5.3 (2.5, 13)	20.3 (9.9, 36.4)	<0.001
∆mTSS ≤ 5 (remission), number (%)	149 (73.0)	121 (90.3)	26 (50.0)	2 (11.1)	<0.001
∆mTSS = 0 (no progression), number (%)	59 (28.9)	55 (41.0)	4 (7.7)	0	<0.001
∆mTSS ≥ 25, number (%)	10 (4.9)	0	2 (3.8)	8 (44.4)	<0.001
Changes in bone erosion for 10 years					
Erosion score at 10 years, mean (SD)	4.4 (6.9)	2.8 (4.9)	7.5 (9.8)	7.5 (6.0)	<0.001
∆erosion score, mean (SD)	1.2 (2.4)	0.4 (1.0)	2.0 (2.9)	4.8 (3.7)	<0.001
∆erosion score ≤ 3, number (%)	180 (88.2)	131 (97.8)	41 (78.8)	8 (44.4)	<0.001
∆erosion score = 0, number (%)	104 (51.0)	85 (63.4)	17 (32.7)	2 (11.1)	<0.001
Changes in joint space for 10 years					
JSN score at 10 years, mean (SD)	9.4 (12.4)	4.9 (7.8)	15.0 (13.0)	26.5 (17.1)	<0.001
ΔJSN score, mean (SD)	4.2 (7.3)	1.3 (1.9)	6.8 (7.2)	18.1 (12.5)	<0.001
ΔJSN score ≤ 3, number (%)	143 (70.1)	105 (91.3)	34 (51.5)	4 (17.4)	<0.001
ΔJSN score = 0, number (%)	71 (34.8)	58 (50.4)	13 (19.7)	0	<0.001

* Differences among the patient groups were assessed using one-way analysis of variance (ANOVA) with Tukey’s HSD post hoc test for continuous variables and Fisher’s exact probability test with the post hoc Holm test for categorical variables. ^†^ Defined as the experience of high or moderate disease activity continuing for ≥12 months (poor control), the experience of high or moderate disease activity continuing for 3–12 months (moderate control), and no experience of high or moderate disease activity continuing for ≥3 months (good control) during 10-year DMARD therapy. ANOVA, analysis of variance; CDAI, clinical disease activity index; DMARD, disease-modifying antirheumatic drug; HAQ-DI, health assessment questionnaire-disability index; HSD, honestly significant difference; IQR, interquartile range; JSN, joint space narrowing; mTSS, van der Heijde-modified total Sharp score; RA, rheumatoid arthritis; SD, standard deviation; ∆, a change from baseline to 10 years later.

**Table 3 jcm-14-06832-t003:** DMARD use in RA patients grouped by structural and functional outcomes after 10-year tight-control therapy.

	Total	Structural Remission ^††^		*p* *	Functional Remission ^††^		*p* *
	(*n* = 204)	Yes	No		Yes	No	
		(*n* = 149)	(*n* = 55)		(*n* = 166)	(*n* = 38)	
MTX use, number (%)	197 (96.6)	142 (95.3)	55 (100)	0.19	161 (97.0)	36 (94.7)	0.62
Use as monotherapy, number (%)	49 (24.0)	41 (27.5)	8 (14.5)	0.065	47 (28.3)	2 (5.3)	0.001
Exposure, months, mean (SD) ^†^	104.3 (28.0)	109.1 (22.7)	92.0 (35.9)	<0.001	107.4 (24.9)	89.7 (37.0)	<0.001
b/tsDMARD use ^‡^, number (%)	155 (76.0)	108 (72.5)	47 (85.5)	0.065	119 (71.7)	36 (94.7)	0.001
Use of 1 class, number (%)	79 (38.7)	61 (40.9)	18 (32.7)	0.33	69 (41.6)	10 (26.3)	0.10
Use of 2 classes, number (%)	40 (19.6)	29 (19.5)	11 (20.0)	1.00	30 (18.1)	10 (26.3)	0.26
Use of ≥3 classes, number (%)	36 (17.6)	18 (12.1)	18 (32.7)	0.001	20 (12.0)	16 (42.1)	<0.001
Number of classes used per patient, mean (SD)	1.4 (1.1)	1.2 (1.0)	1.9 (1.3)	<0.001	1.2 (1.0)	2.3 (1.3)	<0.001
Exposure, months, mean (SD) **^§^**	79.6 (36.1)	76.9 (37.7)	85.8 (31.4)	0.16	77.4 (36.8)	86.8 (32.9)	0.17
Time to the first use, months, mean (SD) **^§^**	23.3 (28.6)	23.7 (29.9)	22.3 (25.4)	0.79	24.0 (29.2)	21.0 (26.4)	0.58
b/tsDMARD failure per patient, number, mean (SD) ^§¶^	0.7 (1.1)	0.5 (1.0)	1.2 (1.3)	<0.001	0.5 (0.9)	1.5 (1.4)	<0.001
No failure, number (%)	95 (61.3)	75 (69.4)	20 (42.6)	0.002	84 (70.6)	11 (30.6)	<0.001
Failure of 1 class, number (%)	26 (16.8)	15 (13.9)	11 (23.4)	0.16	17 (14.3)	9 (25.0)	0.14
Failure of 2 classes, number (%)	19 (12.3)	14 (13.0)	5 (10.6)	0.79	13 (10.9)	6 (16.7)	0.39
Failure of ≥3 classes, number (%)	15 (9.7)	4 (3.7)	11 (23.4)	<0.001	5 (4.2)	10 (27.8)	<0.001

* Compared between patients who achieved remission by 10-year DMARD therapy under tight control and those who failed in achieving remission, using Fisher’s exact probability test for categorical variables and independent-measures *t*-test for continuous variables. ^†^ Determined for 197 patients who received MTX during 10-year DMARD therapy. ^‡^ Including four classes of b/tsDMARDs (TNF inhibitors used for 133 patients [55.4%], IL-6 inhibitors for 70 patients [34.3%], abatacept for 34 patients [16.7%)], and JAK inhibitors for 62 patients [30.4%]). ^§^ Determined for 155 patients who received one or more b/tsDMARDs during 10-year DMARD therapy (108 with structural remission and 47 without it; 119 with functional remission and 36 without it). ^¶^ Defined as failure to achieve remission or low disease activity within 3 months or occurrence of disease flare. ^††^ Defined as ∆mTSS for 10 years ≤ 5.0 for structural remission and HAQ-DI after 10 years ≤ 0.5 for functional remission. DMARD, disease-modifying antirheumatic drug; b/tsDMARD, biological DMARD and targeted synthetic DMARD; HAQ-DI, health assessment questionnaire-disability index; IL-6, interleukin-6; JAK, Janus kinase; mTSS, van der Heijde-modified total Sharp score; MTX, methotrexate; RA, rheumatoid arthritis; SD, standard deviation; TNF, tumor necrosis factor.

**Table 4 jcm-14-06832-t004:** Predictors of structural remission after 10-year DMARD therapy under tight control.

Variables at Baseline	Univariable Analysis		Multivariable Analysis		Multivariable Analysis	
			(Model 1)		(Model 2)	
	Unadjusted OR	*p* *	Adjusted OR	*p* *	Adjusted OR	*p* *
	(95% CI)		(95% CI)		(95% CI)	
Age per additional year	1.02 (0.99–1.05)	0.34	–	–	–	–
Age > 65 years, yes vs. no	1.54 (0.71–3.35)	0.28	–	–	–	–
Male vs. female	2.78 (1.17–6.63)	0.020	3.57 (0.97–13.16)	0.056	3.13 (0.83–11.77)	0.091
Anti-CCP, positive vs. negative	0.33 (0.10–1.16)	0.080	0.35 (0.09–1.43)	0.14	0.30 (0.07–1.29)	0.11
RF, positive vs. negative	0.48 (0.20–1.14)	0.097	–	–	–	–
RF > 200 U/mL, yes or no	0.49 (0.23–1.02)	0.060	0.44 (0.18–1.09)	0.076	0.43 (0.17–1.08)	0.073
DAS28-ESR per additional unit	0.74 (0.57–10.97)	0.030	0.74 (0.53–1.02)	0.066	0.74 (0.53–1.04)	0.083
High (>5.1), yes vs. no	0.62 (0.33–1.15)	0.13	–	–	–	–
Disease duration per additional day	1.00 (1.00–1.00)	0.030	1.00 (1.00–1.00)	0.71	1.00 (1.00–1.00)	0.98
≤6 months, yes vs. no	0.92 (0.49–1.75)	0.81	–	–	–	–
Year of the first visit						
2001–2008	1 (reference)	–	–	–	–	–
2009–2011	1.99 (0.97–4.06)	0.060	2.18 (0.95–5.00)	0.066	2.45 (1.04–5.77)	0.040
2012–2014	2.10 (0.93–4.74)	0.080	1.64 (0.63–4.24)	0.31	1.94 (0.73–5.17)	0.18
Previous csDMARD use, yes vs. no	0.62 (0.28–1.40)	0.25	–	–	–	–
Smoking history > 30 PYs, yes vs. no	2.82 (0.94–8.48)	0.060	2.57 (0.51–12.92)	0.25	2.58 (0.49–13.50)	0.26
BMI > 25, yes vs. no	0.47 (0.22–1.04)	0.060	0.56 (0.22–1.47)	0.24	0.49 (0.18–1.32)	0.16
mTSS						
0 (no erosion/normal joint space)	1 (reference)	–	1 (reference)	–	–	–
>0 and ≤5	0.33 (0.12–0.91)	0.030	0.38 (0.13–1.10)	0.074	–	–
>5 and ≤25	0.16 (0.06–0.43)	<0.001	0.15 (0.05–0.46)	<0.001	–	–
>25	0.09 (0.03–0.29)	<0.001	0.12 (0.03–0.48)	0.003	–	–
Erosion score						
0 (no erosion)	1 (reference)	–	–	–	1 (reference)	–
>0 and ≤3	0.69 (0.32–1.45)	0.32	–	–	1.16 (0.48–2.82)	0.75
>3 and ≤10	0.69 (0.25–1.87)	0.46	–	–	2.18 (0.57–8.37)	0.26
>10	0.19 (0.07–0.52)	0.001	–	–	0.76 (0.19–3.09)	0.70
JSN score						
0 (normal joint space)	1 (reference)	–	–	–	1 (reference)	–
>0 and ≤3	0.63 (0.23–1.72)	0.36	–	–	0.73 (0.23–2.24)	0.58
>3 and ≤10	0.32 (0.13–0.79)	0.014	–	–	0.29 (0.10–0.83)	0.021
>10	0.13 (0.06–0.31)	<0.001	–	–	0.14 (0.04–0.47)	0.002

* Univariable and multivariable logistic regression analyses were conducted to evaluate baseline characteristics associated with structural remission after 10 years of DMARD therapy under tight control. Structural remission was defined as ∆mTSS for 10 years ≤ 5.0. All variables with *p*-values < 0.10 in the univariable models were introduced into multivariable analysis as independent factors using a forced entry procedure. As independent factors, mTSS was introduced into Model 1, and erosion score and JSN score were introduced into Model 2. The multivariable model yielded an AUC–ROC of 0.80 (0.73–0.86, *p* < 0.001) for Models 1 and 0.80 (0.74–0.87, *p* < 0.001) for Model 2. anti-CCP, anti-cyclic citrullinated peptide antibodies; AUC, area under the curve; BMI, body mass index; csDMARDs, conventional synthetic DMARDs; DAS28-ESR, disease activity score 28 joints-erythrocyte sedimentation rate; DMARD, disease-modifying antirheumatic drug; JNS, joint space narrowing; mTSS, van der Heijde-modified total Sharp score; OR, odds ratio; PYs, pack-years; RA, rheumatoid arthritis; RF, rheumatoid factor; ROC, receiver operating characteristic; 95% CI, 95% confidence interval.

**Table 5 jcm-14-06832-t005:** Predictors of functional remission after 10-year DMARD therapy under tight control.

Variables at Baseline	Univariable Analysis		Multivariable Analysis		Multivariable Analysis	
			(Model 1)		(Model 2)	
	Unadjusted OR	*p* *	Adjusted OR	*p* *	Adjusted OR	*p* *
	(95% CIs)		(95% CI)		(95% CI)	
Age per additional year	0.96 (0.92–0.99)	0.015	–	–	–	–
Age > 65 years, yes vs. no	0.33 (0.16–0.70)	0.004	0.29 (0.12–0.70)	0.006	0.27 (0.10–0.69)	0.006
Male vs. female	3.26 (1.10–9.70)	0.034	6.66 (1.91–23.15)	0.003	6.15 (1.66–22.74)	0.006
Anti-CCP, positive vs. negative	0.56 (0.16–1.98)	0.37	–	–	–	–
RF, positive vs. negative	0.85 (0.34–2.08)	0.71	–	–	–	–
RF > 200 U/mL, yes or no	1.27 (0.49–3.31)	0.62	–	–	–	–
DAS28-ESR per additional unit	0.58 (0.42–0.80)	<0.001	0.55 (0.38–0.80)	0.002	0.57 (0.39–0.85)	0.006
High (>5.1), yes vs. no	0.40 (0.19–0.82)	0.013	–	–	–	–
Disease duration per additional day	1.00 (1.00–1.00)	0.24	–	–	–	–
≤6 months, yes vs. no	1.59 (0.78–3.24)	0.20	–	–	–	–
Year of the first visit						
2001–2008	1 (reference)	–	–	–	–	–
2009–2011	1.59 (0.70–3.60)	0.27	–	–	–	–
2012–2014	1.40 (0.57–3.44)	0.46	–	–	–	–
Previous csDMARD use, yes vs. no	0.95 (0.36–2.50)	0.91	–	–	–	–
Smoking history > 30 PYs, yes vs. no	2.37 (0.68–8.24)	0.18	–	–	–	–
BMI > 25, yes vs. no	0.63 (0.26–1.55)	0.32	–	–	–	–
mTSS						
0 (no erosion/normal joint space)	1 (reference)	–	1 (reference)	–	–	–
>0 and ≤5	0.20 (0.055–0.75)	0.017	0.17 (0.04–0.70)	0.014	–	–
>5 and ≤10	0.15 (0.04–0.54)	0.004	0.10 (0.02–0.43)	0.002	–	–
>10	0.08 (0.02–0.32)	<0.001	0.07 (0.01–0.34)	0.001	–	–
Erosion score						
0 (no erosion)	1 (reference)	–	–		1 (reference)	–
>0 and ≤3	0.29 (0.12–0.69)	0.006	–		0.46 (0.17–1.25)	0.13
>3 and ≤10	0.58 (0.16–2.07)	0.40	–		2.00 (0.38–10.54)	0.42
>10	0.17 (0.06–0.53)	0.002	–		0.84 (0.17–4.19)	0.83
JSN score						
0 (normal joint space)	1 (reference)	–	–		1 (reference)	–
>0 and ≤3	0.50 (0.15–1.66)	0.26	–		0.52 (0.14–1.94)	0.33
>3 and ≤10	0.30 (0.10–0.89)	0.029	–		0.17 (0.05–0.58)	0.005
>10	0.12 (0.04–0.30)	<0.001	–		0.09 (0.02–0.38)	<0.001

* Univariable and multivariable logistic regression analyses were conducted to evaluate baseline characteristics associated with functional remission after 10 years of DMARD therapy under tight control. Functional remission was defined as HAQ-DI after 10 years ≤ 0.5. All variables with *p*-values < 0.10 in the univariable models were introduced into multivariable analysis as independent factors using a forced entry procedure. As independent factors, mTSS was introduced into Model 1, and erosion score and JSN score were introduced into Model 2. The multivariable model yielded an AUC–ROC of 0.80 (0.73–0.88, *p* < 0.001) for Model 1 and 0.83 (0.75–0.90, *p* < 0.001) for Model 2. anti-CCP, anti-cyclic citrullinated peptide antibodies; AUC, area under the curve; BMI, body mass index; csDMARDs, conventional synthetic DMARDs; DAS28-ESR, 28-joint disease activity score using erythrocyte sedimentation rate; DMARD, disease-modifying antirheumatic drug; HAQ-DI, health assessment questionnaire-disability index; JSN, joint space narrowing; mTSS, van der Heijde-modified total Sharp score; OR, odds ratio; PYs, pack-years; RA, rheumatoid arthritis; RF, rheumatoid factor; ROC, receiver operating characteristic; 95% CI, 95% confidence interval.

## Data Availability

All data supporting the findings are available from the Human Research Ethics Committee of the NHO Kumamoto Saishun Medical Center for all interested researchers who meet the criteria for access to confidential data. Because these data include patients’ personal information, the Committee does not recommend that such data be made public unnecessarily. Please contact Masahiro Hamaguchi, the Control Manager of the Committee, at 616-syol@mail.hosp.go.jp to request the data.

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
