# Peer review of "Structural and Functional Outcomes in Rheumatoid Arthritis After 10-Year Therapy with Disease-Modifying Antirheumatic Drugs Under Tight Control: Evidence from Real-World Cohort Data"

_jcm, 2025, doi:10.3390/jcm14196832_

Round 1
Reviewer 1 Report
Comments and Suggestions for Authors
10-year outcomes of rheumatoid arthritis are being reported in patients who continued DMARD therapy under tight control (204/572 = 36% of patients). As expected, patients had excellent outcomes. – Possibly a more interesting group would have been those 64% who were excluded.
Traditionally, two clusters of measures have been observed in RA: structural changes cluster with joint deformity and inflammatory activity by lab tests, whereas functional disability clusters with joint tenderness, pain and other patient reported symptoms. Measures within each of the two clusters are correlated with one another, at levels lower than seen within each cluster. Functional disability measures appear more important as they provide more significant predictors of severe outcomes of RA, including work disability, premature mortality, and costs, than radiographs do. (e.g. Pincus et al Best Pract Res Clin Rheumatol. 2007 Aug;21(4):601-28.) Therefore, it is surprising that here, baseline radiographic scores “were strong predictors for physical remission”. This might be opened up in Discussion.
Nomenclature might be aligned: in “conclusion” appears “functional outcomes” but otherwise in text: “physical remission”. “Functional” might be a more familiar to the rheumatology community, to refer to patient self reported measures.
In discussion, more relevant references might be other real-world clinical follow up studies over 10 years, concerning structural and/or functional outcomes, rather than extensions of clinical trials.
I thank the opportunity to comment on your great and extensive work with fine results.
Author Response
Response to Reviewer 1
We are most grateful to Reviewer 1 for their valuable comments and the time and energy spent in reviewing our manuscript. We have made the requested changes and added new information to the manuscript in response to the reviewer’s insightful comments. All alterations are highlighted in red text in the revised manuscript. We are confident that the manuscript has benefited from the reviewer’s useful comments and suggestions.
Below are point-by-point replies to Reviewer 1’s comments.
Comment 1: 10-year outcomes of rheumatoid arthritis are being reported in patients who continued DMARD therapy under tight control (204/572 = 36% of patients). As expected, patients had excellent outcomes. – Possibly a more interesting group would have been those 64% who were excluded.
Author Response: We thank the reviewer for this insightful comment. The aim of this study was to evaluate long-term outcomes and identify baseline characteristics predicting structural and functional remission in rheumatoid arthritis (RA) patients who completed 10-year disease-modifying antirheumatic drug (DMARD) therapy under tight control. Therefore, we first identified 392 patients who started DMARD therapy between April 2001 and July 2014 and had been followed for ≥10 years by July 2024. Then, we excluded 67 patients who dropped out of tightly controlled DMARD therapy during the 10-year follow-up and an additional 121 patients who were not qualified to be included in the present study for other reasons (Figure 1). As the reviewer pointed out, structural damage progression and functional loss are limited in RA patients who completed 10-year tightly controlled DMARD therapy. However, we also showed that, even in these patients, 18 patients (8.8%) experienced high or moderate disease activity lasting for ≥12 months (poor control group), and 52 patients (25.5%) had high or moderate disease activity lasting for 3 to 12 months (moderate control group) (lines 281–287). Structural and functional outcomes were worse in RA patients in the poor control group compared with those who maintained good disease activity control (lines 287–295, Table 2). Considering these data, poor outcomes are expected in RA patients who dropped out of the tight-control approach and those who discontinued DMARD therapy during 10-year follow-up (namely, the patients excluded from this study). We have added these points as one of the limitations in the Discussion section of the revised manuscript (lines 555–563).
Comment 2: Traditionally, two clusters of measures have been observed in RA: structural changes cluster with joint deformity and inflammatory activity by lab tests, whereas functional disability clusters with joint tenderness, pain and other patient reported symptoms. Measures within each of the two clusters are correlated with one another, at levels lower than seen within each cluster. Functional disability measures appear more important as they provide more significant predictors of severe outcomes of RA, including work disability, premature mortality, and costs, than radiographs do. (e.g. Pincus et al Best Pract Res Clin Rheumatol. 2007 Aug;21(4):601-28.)Therefore, it is surprising that here, baseline radiographic scores “were strong predictors for physical remission”. This might be opened up in Discussion.
Author Response: We appreciate the comment regarding the association between disease activity, joint destruction, and functional capability. As described in the Introduction section, joint damage is closely related to physical function in RA patients and health-related quality of life, especially as disease duration increases (lines 42–44). Physical dysfunction has been reported as a consequence of ongoing inflammatory activity and structural joint damage through the course of RA; in early RA, it is most associated with the extent of disease activity, and in late RA, with structural damage. Physical dysfunction is partly reversible, which is mediated by disease activity. Reversibility of physical disability decreases with the duration of RA, and in the absence of disease activity, structural damage contributes to irreversible physical dysfunction. A relationship between structural damage and impaired physical function was also shown in the EURIDISS 10-year follow-up studies. In that study, structural damage and disease activity are independent contributors to impaired physical function early and late in the RA process. In response to the reviewer’s comment, we included this information in the Discussion section (lines 526–535). Four new references have been added (refs. 34 and 50–52).
Comment 3: Nomenclature might be aligned: in “conclusion” appears “functional outcomes” but otherwise intext: “physical remission”. “Functional” might be a more familiar to the rheumatology community, to refer to patient self-reported measures.
Author Response: We acknowledge the reviewer’s comment on this point. In response to this comment, we replaced the terms “physical outcome” and “physical remission” with the terms “functional outcome” and “functional remission” throughout the manuscript; namely, title, abstract, text, tables, and figures.
Comment 4: In discussion, more relevant references might be other real-world clinical follow up studies over 10 years, concerning structural and/or functional outcomes, rather than extensions of clinical trials.
Author Response: We thank the reviewer for this comment on the publications we cited in this study. In the Discussion section, we wanted to compare the structural and functional outcomes in our study with those obtained from previous studies in which RA patients received DMARD therapies over 10 years under tight control. As mentioned in the Introduction section, however, there is limited information regarding long-term protective effects of the tight-control approach on structural damage progression and functional loss, especially after various biological and non-biological targeted DMARDs have become available in RA treatment (lines 72–76). Regarding this topic, several clinical trials and their extension studies are available, which we have introduced in the Discussion section (the Best, IMPROVED study, PREMIER, DE019, and ASPIRE studies). Regarding real-world follow-up studies over 10 years, we have cited 5 publications that addressed joint damage progression and physical dysfunction (refs. 31–35). Refs. 34 and 35 are newly included in the revised version of the manuscript. Most patients in these studies were treated with MTX and other csDMARDs without tight control during follow-up. Their structural and functional outcomes were worse compared with our patients (lines 417–426). In a systematic review and meta-analysis using data from longitudinal observational cohorts (follow-up range, 5–20 years), Carpenter et al. showed that the annual progression rate of structural joint damage in studies recruiting between 1990 and 2011 was significantly lower than that in studies recruiting between 1965 and 1989 because of critical changes in treatment practices. We added information from this systematic review and meta-analysis to the Discussion section of the revised manuscript (lines 426–431), and a new reference was added (ref. 36). Additionally, we have cited 3 real-world cohort studies with 10-year follow-up addressing predictive factors for joint damage progression in the Discussion section (refs. 31, 33, and 47; lines 508–511 and 543–547). Last, we cited a 10-year prospective cohort study to discuss the effect of disease activity control on radiographic damage and functional loss (lines 479–483; ref. 44). In the real-world setting, frequent and regular monitoring of disease activity as well as early switching and adjustment of DMARD therapy over 10 years might not be feasible. However, we believe that this approach is beneficial to favorable structural and functional outcomes in RA patients treated in routine clinical practice.

Reviewer 2 Report
Comments and Suggestions for Authors
Review article: Structural and physical outcomes in rheumatoid arthritis after 10-year therapy with disease-modifying antirheumatic drugs under tight control: evidence from real-world cohort data
An interesting paper discusses the structural and physical outcomes in rheumatoid arthritis after 10-year therapy with disease-modifying antirheumatic drugs.
Some revisions are needed.
- The abstract is correctly written.
- Keywords: correctly
- Section: Introduction: Please describe in more detail how tight-control strategy improves clinical and radiographic outcomes. What exists in the literature?
- Section: materials and methods: Please explain why you excluded patients who had discontinued DMARDs for one month. What about patients who, for example, temporarily stopped therapy due to infections, elevated liver enzymes, surgical interventions, or other reasons?
- Section Results : is correctly written.
Table is correctly. There is a 'year of first visit' column in the tables. Could you explain whether there were any differences in the investigated parameters based on the year of first visit? Specifically, are there any differences between the periods 2001–2008, 2009–2011, and 2012–2024?
- Section: Discussion: Regardless of what you wrote — 'We could not examine the effects of each DMARD therapy on long-term outcomes because of the small number of patients in individual treatment groups and the various exposure times' — I believe it would still be interesting to explore the effects of individual DMARD therapies, particularly the switching from one bDMARD to another. I suggest adding a few sentences to the discussion to elaborate on this aspect.
- Conclusion : correctly written.
- Section Reference is correctly written.
Author Response
Response to Reviewer 2
We are most grateful to Reviewer 2 for their valuable comments and the time and energy spent in reviewing our manuscript. We have made the requested changes and added new information to the manuscript in response to the reviewer’s insightful comments. All alterations are highlighted in red text in the revised manuscript. We are confident that the manuscript has benefited from the reviewer’s useful comments and suggestions.
Below are point-by-point replies to Reviewer 2’s comments.
Comment 1: The abstract is correctly written.
Comment 2: Keywords: correctly
Comment 7: Conclusion : correctly written.
Comment 8: Section Reference is correctly written.
Author Response: We greatly appreciate the reviewer’s evaluation on these points.
Comment 3: Section: Introduction: Please describe in more detail how tight-control strategy improves clinical and radiographic outcomes. What exists in the literature?
Author Response: We thank the reviewer for the comment regarding the tight-control approach to the management of patients with rheumatoid arthritis (RA). Tight control is a treatment strategy tailored to individual patients with the aim of achieving a predefined level of disease activity within a certain period of time. This treatment strategy is attained by careful and regular monitoring of disease activity using validated composite measures, as well as early therapeutic adjustments or switches of disease-modifying antirheumatic drug (DMARD) therapies that fail to adequately control disease activity. In previous clinical trials, the treatment goal was clinical remission or low disease activity; measurements of disease activity were performed every 1 to 3 months and, until the desired treatment goal was reached, DMARD therapies were adjusted at least every 3 months. These trials showed that compared with conventional approaches, the tight-control strategy adequately controlled clinical disease activity and substantially reduced radiographic progression in RA patients. We added this information to the Introduction section (lines 53–65; refs. 9–16).
Comment 4: Section: materials and methods: Please explain why you excluded patients who had discontinued DMARDs for one month. What about patients who, for example, temporarily stopped therapy due to infections, elevated liver enzymes, surgical interventions, or other reasons?
Author Response: We greatly appreciate this insightful comment. The aim of this study was to evaluate long-term outcomes and predictors of structural and functional remission in RA patients who completed 10-year DMARD therapy under tight control. Therefore, we excluded RA patients who had discontinued DMARD therapy during the 10-year follow-up. As shown in Figure 1, the excluded cases included 17 patients who discontinued DMARD therapies due to adverse events such as infection (n = 4), fracture (n = 4), malignancy (n = 3), Fabry disease (n = 1), dementia (n = 1), de novo hepatitis B virus infection (n = 1), cardiac infarction (n = 1), chronic pulmonary embolism (n = 1), and lymphopenia (n = 1), as well as 12 patients who discontinued DMARD therapy based on patient’s preference (e.g., economic reason, fear of adverse events). These patients did not restart DMARD therapies during follow-up. As the reviewer pointed out, we did not exclude patients who discontinued DMARD therapies temporarily for certain reasons (e.g., liver enzyme elevations, infection, surgical interventions) and restarted them within 1 month. In the previous version of the manuscript, we intended to emphasize this point; that is, we included the discontinuation cases within 1 month, but our description was unclear. To clarify this point, we have revised Figure 1 and the Materials and Methods section (lines 104–106, 118, and 119). We thank the reviewer again for pointing out this critical point.
Comment 5: Section Results : is correctly written. Table is correctly. There is a 'year of first visit' column in the tables. Could you explain whether there were any differences in the investigated parameters based on the year of first visit? Specifically, are there any differences between the periods 2001–2008, 2009–2011, and 2012–2024?
Author Response: We appreciate this important comment. As the reviewer pointed out, we used year of patient’s first visit as a baseline characteristic. The availability of DMARDs differed between 2001 and 2014. From 2001 to 2008, TNF inhibitors were the only available biological DMARD (bDMARD) in Japan. Subsequently, interleukin-6 inhibitors and abatacept became commercially available between 2009 and 2011, and the use of Janus kinase (JAK) inhibitors began in 2012. The aim was to explore the possibility that differences in DMARD availability during the early stage of a patient’s treatment might contribute to outcomes after 10 years. There were no significant differences in structural or functional remission rates by year of patient’s first visit. In addition, the year of first visit between 2012 and 2014 (in that period, all classes of bDMARDs and JAK inhibitors were available) was not a significant predictor for structural or functional remission (Tables 1, 4, and 5). We added an explanation of why we used year of first visit as a baseline characteristic to the Materials and Methods section (lines 135–138) and the Table 1 footnote (lines 225 and 226) in the revised manuscript.
Comment 6: Section: Discussion: Regardless of what you wrote – 'We could not examine the effects of each DMARD therapy on long-term outcomes because of the small number of patients in individual treatment groups and the various exposure times' – I believe it would still be interesting to explore the effects of individual DMARD therapies, particularly the switching from one bDMARD to another. I suggest adding a few sentences to the discussion to elaborate on this aspect.
Author Response: Thank you very much for the suggestion to add more discussion concerning the effect of switching from one bDMARD/targeted synthetic DMARD (tsDMARD) to another on long-term outcomes in RA patients. As shown in Table 3, RA patients in the non-remission groups received more classes of bDMARD/tsDMARD and experienced more failures during 10-year DMARD therapy compared with those in the structural remission and functional remission groups. The rate of users of ≥3 bDMARD/tsDMARD classes was significantly higher in the non-remission groups compared with the remission groups. Users of ≥3 classes had 2 or more switches of a current bDMARD/tsDMARD to another one mainly because of the need for treatment adjustments to control clinical disease activity. In other words, these patients might experience high or moderate disease activity for a longer time than users of 1 or 2 bDMARD/tsDMARD classes. The data support the idea that sustained remission or at least low disease activity during follow-up is important in achieving structural and functional remission after 10 years. The overriding principle of the tight-control approach is not the choice of individual DMARD therapy but rather the regular monitoring and early switching or adjustments of DMARD therapy to achieve good control of disease activity. In the present study, we assessed clinical disease activity every 3 months and, if at least low disease activity was not achieved within 3 months, DMARD therapy was modified or changed. However, more frequent monitoring and earlier switching or adjustments of DMARD therapies may be required for patients with high or moderate disease activity. We have added this discussion to the revised manuscript (lines 491–506).

Round 2
Reviewer 2 Report
Comments and Suggestions for Authors
no comments and suggestion for Authors